# Revisiting "Edit Away and My Face Will not Stay: Personal Biometric Defense against Malicious Generative Editing"

**Luis Vitor Zerkowski**      *luisvz@gmail.com*
*University of Amsterdam*

**Soham Chaudhuri**      *sohamchaudhuri25@gmail.com*
*University of Amsterdam*

**Finley Helms**      *finletastic@gmail.com*
*University of Amsterdam*

**Jelle Sombekke**      *jellesombekke@gmail.com*
*University of Amsterdam*

**Udit Thakur**      *udit.thakur@student.uva.nl*
*University of Amsterdam*

**Reviewed on OpenReview:** *https://openreview.net/forum?id=5Q1gr8OAXU*

## Abstract

Recent advances in diffusion-based image editing have enabled highly realistic and accessible manipulation of facial images, raising serious concerns about biometric privacy and malicious misuse. FaceLock, introduced in Edit Away and My Face Will Not Stay: Personal Biometric Defense against Malicious Generative Editing, proposes an optimization-based defense that embeds subtle perturbations into images at publication time to induce identity distortion in downstream generative edits. The method claims prompt-agnostic effectiveness and strong performance across multiple editing scenarios, supported by open-source code. In this paper, we present a systematic reproducibility study of FaceLock that evaluates its technical, quantitative, and qualitative reproducibility. We assess whether the reported results can be obtained using the released codebase, analyze the correspondence between the paper's algorithmic description and its implementation, and document ambiguities that impact reproducibility. We further examine quantitative reproducibility by attempting to recover the reported performance trends and relative ranking against baselines. We, however, were not able to reproduce the originally reported performance trends, and our outputs were generally worse than those presented in the original paper. Beyond that, we expand the qualitative analysis to a broader set of image–prompt pairs and an additional, harder facial dataset to better test generalization behavior. While we obtained some successful outputs, only a small fraction of our qualitative results matched the consistently high quality reported by the authors. Finally, we introduce an extension to the FaceLock method that helps with robustness, and we critically examine the evaluation criteria used to measure defense effectiveness, highlighting limitations of prompt fidelity as a primary metric and arguing for a more explicit consideration of the trade-off between identity protection and preservation of the original image. We provide a link to our GitHub repository [1].

---

[1]https://github.com/Luizerko/revisiting_facelock

# 1 Introduction

In recent years, rapid progress in generative models for image editing, particularly diffusion-based methods, has enabled increasingly realistic transformations through natural language prompts, allowing facial attributes, expressions, and even semantic context to be altered with minimal effort and high visual fidelity (Brooks et al. (2022); Shuai et al. (2024)). While these advances unlock powerful creative applications, they also introduce significant privacy concerns when applied to images of real individuals. The ease with which facial images can be manipulated raises the risk of malicious misuse, including identity alteration, impersonation, non-consensual content creation, and misinformation (Korshunov & Marcel (2018); Tolosana et al. (2020)). Once an image is publicly released, its owner typically has little control over how it may be edited by third parties using powerful generative tools, creating an asymmetry between image ownership and editing capability that motivates the need for defensive mechanisms to protect biometric identity while naturally preserving the ability to share images online.

In response to these concerns, recent work has proposed biometric defense systems that modify images in subtle ways to disrupt downstream generative editing (Chen et al. (2023); Choi et al. (2024); Salman et al. (2023)). By embedding carefully designed perturbations at the time of image publication, some approaches aim to maintain visual fidelity of images while preventing unauthorized transformations, while others target complete image distortion, always without relying on platform-level enforcement. Within this paradigm, the paper "Edit Away and My Face Will Not Stay: Personal Biometric Defense against Malicious Generative Editing" (Wang et al. (2024)) introduces *FaceLock*, an optimization-based defense pipeline designed to operate under these principles. The work is particularly notable because, in contrast to much of the prior literature, it frames defense in terms of inducing biometric distortion in the edited output rather than making the protected image difficult to edit or completely deforming it. The authors report strong quantitative and qualitative performance across multiple metrics and editing scenarios, provide an open-source implementation, and evaluate their method on Celeba-HQ (Karras et al. (2017)), a publicly available dataset. Taken together, the methodological novelty, strong empirical claims, and open-source accessibility makes FaceLock a compelling candidate for reproducibility analysis.

Given these considerations, this study contributes with:

- A comprehensive audit of FaceLock, assessing the validity of its core claims from both quantitative and qualitative perspectives. Our findings suggest that, in practice, the defense's performance may be less effective than originally reported and may also contrast with the high reliability implied by the original work.

- A FaceLock analysis expansion to a broader range of image-prompt pairs and a more challenging facial dataset to better characterize the method's limitations.

- A more rigorous evaluation framework for assessing defenses in generative-editing contexts. Our proposed pipeline explicitly balances biometric protection with image fidelity prior to editing, providing a more transparent standard for the field. Under this evaluation, FaceLock leans towards minimal image distortion and compromised editing protection, but is still surpassed by a baseline that offers both lower distortion and superior protection.

- And an extension to FaceLock designed to improve robustness against the stochasticity in diffusion-based editing. Evaluations show measurable, though modest, performance gains.

# 2 Scope of Reproducibility

The objective of this study is to systematically evaluate the claims presented in Wang et al. (2024) across multiple dimensions and, grounded in our reproducibility findings, contribute meaningful advancements for the field. We begin with a technical assessment to verify clarity across the entire workflow, starting all the way from the provided installation instructions, dataset preparation and configuration of hyperparameters. This phase also includes a close inspect of the fidelity between the paper's algorithmic descriptions and the released implementation to determine if the codebase accurately reflects the described method. Our quantitative and

qualitative analyses then examine whether FaceLock's reported performance trends hold under reproduction. Specifically, we test for the replication of absolute metrics, the relative ranking of FaceLock against proposed baselines, and the consistency of qualitative behavior relative to the visual examples presented in the original work.

To take reproducibility a step further, we assess generalization by expanding the scope of experiments to the challenging FaceForensics++ dataset (Rössler et al. (2019)), moving beyond the high-quality CelebA-HQ (Karras et al. (2017)) and Flickr-Faces-HQ (Karras et al. (2019)) distributions used in the original research. We then conclude with a methodological critique and a few extensions. First, we examine the suitability of existing metrics for measuring defense effectiveness, advocating for context-sensitive evaluation tailored to specific defense objectives. Additionally, we propose a new framework that explicitly captures the trade-off between biometric protection and image fidelity in publication time. Finally, we introduce a modification to the FaceLock algorithm designed to mitigate its sensitivity to the inherent randomness of diffusion-based editing. In doing so, we traverse the entire reproducibility cycle: from implementing and assessing the original work to identifying its limitations and proposing complementary improvements to both the method and the broader field.

## 3 Methodology

### 3.1 Problem Setup

The authors of Wang et al. (2024), from now on referred to as **the Authors**, consider the problem of protecting an individual's biometric identity against malicious generative image editing. Given a facial image released publicly by its owner, the threat model assumes that an adversary may apply arbitrary diffusion-based editing methods with unconstrained natural language prompts. Different from most previous work in the field, the goal of the defense is not to prevent editing itself or completely distort the image, but to ensure that whatever the edited output looks like, it no longer preserves the subject's biometric identity. Importantly, the defense must operate prior to image publication, without access to the adversary's prompts or editing model. Following that, there's an unmeasured expectation that the protected image should remain visually natural and as close as possible to the original image so as not to affect sharing it instead of the original one.

Under this formulation, the defense produces a protected image by adding an optimization-computed subtle perturbation to the original input. The Authors define success in terms of low prompt fidelity and identity distortion in the edited results, which we'll further discuss in later sections. Specifically, after a protected image is edited, the resulting output should not follow what the prompt required and be dissimilar to the original identity in facial recognition embedding space. This framing shifts the objective from edit obstruction to biometric disruption, although they still account for low prompt fidelity, and emphasizes robustness across a wide range of prompts rather than protection against a predefined set of transformations.

### 3.2 Diffusion Model

All generative editing experiments in this work rely on pretrained diffusion-based image editing models. These models are used both to test a defense pipeline implementation via editing and within some defense pipelines themselves. The Authors use InstructPix2Pix (Brooks et al. (2022)), a model that operates using latent diffusion and is publicly available through the Hugging Face Diffusers library (von Platen et al. (2022)). The released model checkpoint [2] is fully pretrained and is used as-is in both the Authors' and our experiments.

### 3.3 Defense Models

We evaluate three image-level biometric defense methods: FaceLock, which is the primary focus of our reproducibility study, and two main defenses the Authors used as baselines, PhotoGuard (Choi et al. (2024)) and EditShield (Chen et al. (2023)). All three methods compute image-specific perturbations prior to the

---

[2]https://huggingface.co/timbrooks/instruct-pix2pix

image editing process, but they differ substantially in their optimization objectives and their reliance on the underlying generative model.

### 3.3.1 PhotoGuard

PhotoGuard is a diffusion-based image protection method that generates adversarial perturbations via gradient-based optimization in the latent space of a diffusion model. Unlike approaches relying on implicit guidance, it explicitly differentiates through the generative pipeline by optimizing the input through the VAE encoder (Kingma & Welling (2022)) of Stable Diffusion, back-propagating gradients through the encoding process to manipulate the latent representation. Concretely, PhotoGuard applies projected gradient descent to update the input image by minimizing the distance between its latent representation and a predefined target latent, while constraining the perturbation using an $\ell_\infty$ budget and spatial masking to restrict changes to selected regions and keep pixel values valid. Because the perturbations are tightly coupled to diffusion internals, the method requires white-box access to the VAE encoder.

### 3.3.2 EditShield

EditShield is an image-level defense against instruction-guided diffusion-based editing that disrupts the image conditioning signal by perturbing the latent representation used to guide the diffusion process. Unlike FaceLock, it does not aim to distort identity, but instead interferes with editing itself by optimizing perturbations so that the VAE latent of the protected image deviates from that of the original image, reducing the effectiveness of downstream edits. Formally, EditShield maximizes a latent inconsistency loss between the VAE-encoded latents of the original and protected images, while preserving visual fidelity through an $\ell_2$ image-space regularizer. To improve robustness against common transformations, it further applies Expectation over Transformation (Athalye et al. (2018)), optimizing the latent inconsistency over a distribution of editing conditions while keeping the generation pipeline fixed.

### 3.3.3 FaceLock

FaceLock is an image-level biometric protection method designed to disrupt identity preservation in diffusion-based image editing models. Given an input image, FaceLock computes a bounded adversarial perturbation by optimizing directly in pixel space under an $\ell_\infty$ constraint. Rather than attempting to prevent edits from occurring or completely messing them up, the method aims to ensure that any edited output no longer preserves the original subject's biometric identity. The optimization objective combines two terms that balance identity distortion against visual fidelity. Specifically, it includes a face recognition loss that maximizes identity discrepancy between the protected and original images using a pretrained face recognition model, and a perceptual similarity loss based on LPIPS (Zhang et al. (2018)) to preserve visual fidelity.

FaceLock relies on pretrained face recognition and alignment models to define its identity-based optimization objective. Identity embeddings are computed using the ViT-KPRPE architecture (Kim et al. (2024)), trained with the AdaFace loss (Kim et al. (2023)) on the WebFace4M dataset [3] (Zhu et al. (2021)) available on Hugging Face). Prior to identity comparison, facial regions are extracted and aligned using a DFA-MobileNet face alignment model [4] (Kim et al. (2024)). These models are used exclusively during the defense optimization process to guide perturbation generation and are not involved in the downstream image editing stage.

### 3.3.4 FaceLock Extension: EoDR FaceLock

We introduce Expectation over Diffusion Randomness (EoDR) FaceLock, an extension of FaceLock designed to improve robustness to the stochasticity of diffusion-based editing. While the name is inspired by Expectation over Transformation from Athalye et al., our approach does not apply explicit transformations to the input image. Instead, it targets a failure mode that we observe and analyze in more detail later in the qualitative results section: a high sensitivity to the random seed used during editing. This behavior is particularly evident in Figure 8, where we attempt to reproduce the qualitative examples from Figure 6 of

---

[3]https://huggingface.co/minchul/cvlface_adaface_vit_base_kprpe_webface4m
[4]https://huggingface.co/minchul/cvlface_DFA_mobilenet

Wang et al. (2024). We observe that, for a given image, FaceLock may succeed for some seeds, but fails for most of them, indicating that a key shortcoming of the method is its lack of robustness to randomness.

Hence, as the name suggests, the core idea of our extension is to optimize the adversarial perturbation under an expectation over diffusion randomness, rather than for a single deterministic editing trajectory. Concretely, we simulate stochastic diffusion behavior during defense optimization by injecting scheduler-consistent noise into the VAE latent of the adversarial image at randomly sampled timesteps from the later denoising stages. We also replace the deterministic use of the latent mean with a (reparameterized) latent sample, introducing additional stochasticity. The defense loss is then computed as the average face-recognition loss over multiple such noisy latent samples, and gradients are taken through this expectation. This encourages the protected image to remain identity-disruptive under different diffusion noise realizations, improving robustness across seeds. The formal procedure for this extension is detailed in Algorithm 1.

Algorithm 1: EoDR FaceLock.

**Inputs:** original image $x$; adversarial image at $i$th iteration $x^i$; VAE encoder $E$, decoder $D$; scheduler noise function of diffusion editing pipeline $\mathcal{N}(\cdot, t)$; face recognition model $f_{\mathrm{FR}}$; face recognition loss $\mathcal{L}_{\mathrm{FR}}$; EoDR samples $K$; late diffusion timestep range $\mathcal{T}$.
**Output:** protected image $\tilde{x}$.
**Procedure:**

1. Set $\ell \leftarrow 0$
2. For $k = 1, \ldots, K$:
   (a) Sample timestep $t \sim \mathcal{T}$
   (b) Encode latent distribution $(\mu, \sigma) \leftarrow E(x^i)$
   (c) Sample (reparameterized) latent $z \leftarrow \mu + \sigma \odot \eta$
   (d) Add scheduler-consistent noise $\hat{z} \leftarrow \mathcal{N}(z, t)$
   (e) Decode $\hat{x}^i \leftarrow D(\hat{z})$
   (f) Accumulate $\ell \leftarrow \ell + \mathcal{L}_{\mathrm{FR}}\big(f_{\mathrm{FR}}(\hat{x}^i), f_{\mathrm{FR}}(x)\big)$
3. Average $\ell \leftarrow \ell / K$

## 3.4 Datasets and Processing

The Authors primarily evaluate FaceLock on CelebA-HQ (Karras et al. (2017)), providing a download link and reporting experiments on 2000 images, though the paper does not specify the split used. Since the CelebA-HQ validation set contains exactly 2000 images (compared to 28,000 training images), we infer that evaluation was likely performed on the validation split and we ran our experiments on the entirety of the presumed dataset.

A key reproducibility issue arises from the mismatch between the dataset resolution ($1024 \times 1024$) and the released FaceLock implementation, which is configured for $512 \times 512$ inputs. The codebase contains no resizing logic, despite directing users to a higher-resolution dataset, and provides little documentation on this issue. The expected input resolution is mentioned only once in Wang et al. (2024), in Table A2 of the appendix, forcing users to infer the correct setup and implement their own resizing procedure. Running the pipeline at the full dataset resolution requires manual code modifications and, even when attempted, leads to crashes caused by numerical overflows. Therefore, to execute the pipeline reliably and in accordance with the originally described experiments, we additionally resize the inputs to $512 \times 512$ using bicubic interpolation.

### 3.4.1 Additional Dataset: FaceForensics++

To assess the generalizability of FaceLock beyond the CelebA-HQ and the additional Flickr-Faces-HQ (Karras et al. (2019)) mentioned in the appendix of Wang et al. (2024), we extend our evaluation to FaceForensics++ (Rössler et al. (2019)), a large-scale benchmark for facial manipulation detection that contains real-world and manipulated online videos under varying compression levels. Compared to CelebA-HQ and Flickr-Faces-HQ, which consist of high-quality, well-centered face images, FaceForensics++ is more challenging due to lower visual fidelity, motion blur, compression artifacts, and higher variability in pose, illumination, and

background, with faces not always centered and sometimes multiple faces per frame. We experiment on the 1000 benchmark frames and, to align with the released FaceLock implementation, we preprocess each image by detecting faces and cropping around the largest detected face (when multiple faces were present), then resize all images to $512 \times 512$ using bicubic interpolation.

### 3.5 Experimental Setup

Having discussed the datasets and associated preprocessing, we now describe the experimental setup used in our reproduction study and highlight inconsistencies encountered in the released implementation. We first edited the original images using the 25 prompts provided by the Authors, spanning three categories - facial feature modifications, accessory adjustments, and background alterations - to cover a broad range of instruction-guided editing scenarios. We then defended the original images with PhotoGuard, EditShield and FaceLock, edited the defended images using the same prompts, and evaluated the resulting outputs. To reduce stochastic effects, we followed the Authors' recommended procedure of keeping the editing process consistent between original and defended images, including the same seed. For all editing pipelines, we followed the hyperparameter configurations reported in the appendix of the original paper (Table 1).

Table 1: Editing pipeline configuration used in our experiments, following the appendix hyperparameter settings from the original paper.

| Editing Model | Inference Steps | Image Guidance Scale | Text Guidance Scale | Random Seed |
|---|---|---|---|---|
| Instruct-Pix2Pix | 50 | 1.5 | 7.5 | 42 |

For the defense pipelines, in turn, we used the hyperparameters reported by the Authors for FaceLock (Table 2), but left the default defense parameters as they were for the baselines since the they never mention these parameter in Wang et al. (2024).

Table 2: FaceLock defense configuration used in our experiments, following the appendix hyperparameter settings from the original paper.

| Perturbation Budget ($\epsilon$) | Step Size ($\alpha$) | Optimization Steps ($N$) | Weighting Parameter ($\lambda$) |
|---|---|---|---|
| 0.02 | 0.003 | 100 | 0.2 |

While we executed the released implementation as provided to avoid introducing additional deviations, we observed multiple discrepancies between the paper and the code that may affect reproducibility. First, Algorithm 1 in the appendix states that the perturbation should be initialized from a normal distribution $\mathcal{N}(0, \mathbf{I})$. In contrast, the released code initializes the perturbation by sampling from a uniform distribution $\mathbf{U}(-\epsilon, \epsilon)$. This difference in initialization is not documented and may influence optimization behavior.

More critically, the optimization objective implemented in the code does not seem to correspond to Equation (5) of the paper, which defines the core FaceLock method. In the paper, $f_{\mathrm{FE}}$ is weighted by $\lambda$, whereas in the implementation this term (`loss_lpips`) is not multiplied by $\lambda$ at all. Instead, the weighting parameter is applied to a regularization term (`loss_encoder`) that is not described anywhere in the paper. Moreover, the feature embedding loss is only activated after the first 25% of optimization steps, and the identity distortion loss $f_{\mathrm{FR}}$ (`loss_cvl` in the code) is only enabled after 35% of the iterations. In summary, the loss computation that would theoretically lead to the perturbation $\delta$ is rather implemented as:

$$\mathrm{loss} = -\mathrm{loss}_{\mathrm{cvl}} \cdot \mathbf{1}_{i \geq 0.35N} + 0.2 \cdot \mathrm{loss}_{\mathrm{encoder}} + \mathrm{loss}_{\mathrm{lpips}} \cdot \mathbf{1}_{i > 0.25N}$$

We ratify that this staged optimization strategy is neither described nor motivated in the paper, despite Equation (5) being presented as the central methodological contribution, and it resembles the EditShield optimization objective with a few tweaks.

As for EoDR FaceLock, we aimed to eliminate several of the arbitrary design choices we discussed. While we found the regularizer empirically important and therefore retained it, we removed the heuristic conditions

controlling when `loss_cvl` and `loss_lpips` begin to take effect, and introduced an explicit weighting factor $\rho$ for the $f_{\text{FE}}$ term. Since this variant is explicitly designed for robustness, we also observed that the loss converges in substantially fewer iterations. All differences are summarized in Table 3.

Table 3: EoDR FaceLock defense configuration changes from vanilla FaceLock.

| Loss Objective | Optimization Steps ($N$) | Regularizer Weight ($\lambda$) | Image Feat. Ext. Weight ($\rho$) |
|---|---|---|---|
| $-\text{loss}_{\text{cvl}} + \lambda \cdot \text{loss}_{\text{encoder}} + \rho \cdot \text{loss}_{\text{lpips}}$ | 30 | 0.2 | 0.8 |

Finally, the feature embedding disparity is computed using the LPIPS model in the implementation, even though LPIPS is introduced in the paper solely as an evaluation metric. Since FaceLock achieves strong LPIPS scores in the reported experiments, we believe that this design choice is important to disclose: the performance on LPIPS is not an emergent property of the method, but rather a direct consequence of optimizing for it. While optimizing for a given metric is not inherently problematic, the lack of transparency may lead readers to misinterpret the results.

Now to evaluate our experimental setup, we initially follow the methodology outlined in the original paper, assessing two main factors: prompt fidelity, which quantifies the disruption caused by editing through CLIP-S (Radford et al. (2021)), PSNR, SSIM (Wang et al. (2004)) and LPIPS scores, and image integrity, which measures how closely the edited image resembles the original through CLIP-I and FR scores. Once reproducibility is done, we repurposed the prompt fidelity metrics to assess image integrity between the original and defended images, an additional evaluation pipeline we better discuss in the next section.

## 4 Results

We now present the results of our evaluation from both quantitative and qualitative perspectives. The quantitative analysis provides a numerical comparison of FaceLock's (Wang et al. (2024)) performance across multiple metrics, focusing on its relative behavior compared to baseline methods and highlighting that we were unable to reproduce most performance trends reported by the Authors. The qualitative analysis offers complementary insights into the model's behavior, capturing nuanced failure modes that are not always reflected in aggregate metrics and that, in several cases, differ from the interpretation presented in the original paper.

### 4.1 Quantitative Results

As shown in Table 4, we were unable to reproduce some key results reported in the original paper. This discrepancy is evident already in the absolute metric values. While our results for the baseline methods remain within the standard deviation of those reported by the Authors, the results for FaceLock differ substantially. Moreover, the discrepancy also affects the reproducibility of the reported performance trends, particularly the relative ranking among methods.

In our experiments, EditShield (Chen et al. (2023)) generally achieves better scores than the other methods. By contrast, both FaceLock and the version of PhotoGuard (Choi et al. (2024)) used in our experiments, consistent with the authors' original setup, performed a bit worse, with FaceLock consistently underperforming PhotoGuard in our evaluation. As a result, we are unable to confirm the Authors' claim that FaceLock provides state-of-the-art defense performance.

PhotoGuard and EditShield achieve better prompt fidelity scores than FaceLock, but we view this difference as expected and not particularly informative. The former are designed to either suppress or distort editing, whereas FaceLock prioritizes removing biometric identity regardless of prompt adherence. This highlights a limitation in the evaluation protocol: prompt fidelity is an ill-suited metric for comparing defenses with such fundamentally different objectives. Despite this, the Authors report unexpectedly strong prompt fidelity for FaceLock, achieving the best SSIM (Wang et al. (2004)) and LPIPS (Zhang et al. (2018)) and staying close to baselines on other metrics, which we found surprising and were unable to reproduce. In our experiments,

Table 4: Quantitative evaluation on prompt fidelity (CLIP-S, PSNR, SSIM, LPIPS) and image integrity (CLIP-I, FR). Arrows (↑ or ↓) indicate whether a higher or lower value is preferred for a successful defense. All results are averaged over the 25 prompts for editing. *Simple FaceLock* is not taken into consideration for the numerical ranking between methods as it is a deliberately ill-posed baseline, intended only to demonstrate the necessity of our proposed additional evaluation framework.

| Method | Prompt Fidelity | | | | Image Integrity | |
|---|---|---|---|---|---|---|
| | CLIP-S ↓ | PSNR ↓ | SSIM ↓ | LPIPS ↑ | CLIP-I ↓ | FR ↓ |
| PhotoGuard | $0.1128 \pm 0.0316$ | **$17.83 \pm 1.74$** | $0.6489 \pm 0.0450$ | $0.3830 \pm 0.0427$ | $0.7539 \pm 0.1002$ | $0.7067 \pm 0.2115$ |
| EditShield | **$0.1075 \pm 0.0275$** | $18.31 \pm 2.35$ | **$0.5663 \pm 0.0746$** | **$0.4616 \pm 0.0503$** | **$0.7154 \pm 0.1044$** | **$0.6051 \pm 0.2336$** |
| FaceLock | $0.1139 \pm 0.0372$ | $25.26 \pm 2.13$ | $0.7849 \pm 0.0382$ | $0.2438 \pm 0.0264$ | $0.8495 \pm 0.0737$ | $0.8387 \pm 0.1112$ |
| EoDR FaceLock | $0.1133 \pm 0.0375$ | $25.45 \pm 2.11$ | $0.7911 \pm 0.0368$ | $0.2376 \pm 0.0250$ | $0.8502 \pm 0.0730$ | $0.7809 \pm 0.1014$ |
| Simple FaceLock | $0.0997 \pm 0.0191$ | $15.81 \pm 1.60$ | $0.6280 \pm 0.0844$ | $0.3832 \pm 0.0840$ | $0.5694 \pm 0.0621$ | $0.2502 \pm 0.1219$ |

FaceLock behaved more consistently with its stated identity-focused design, reinforcing our view that prompt fidelity gives a weak perspective for comparing the defenses.

Turning to image integrity, unlike prompt fidelity metrics, which we argued are poorly aligned with Face-Lock's objective, these metrics are more relevant for evaluating FaceLock, and we therefore view their inclusion as a valuable step by the Authors. However, FaceLock's performance does not follow the trends reported in the original paper and instead yields the worst results among baselines. Notably, FaceLock produces image integrity scores that are even worse than PhotoGuard, despite PhotoGuard being primarily designed for edit prevention and typically incurring low integrity only when it fails. This suggests that the proposed pipeline is unreliable and makes it difficult to determine when FaceLock meaningfully influences editing, a point we further substantiate in our qualitative analysis. Consequently, we were unable to confirm the Authors' claim that FaceLock achieves state-of-the-art performance even under their own proposed image integrity evaluation pipeline.

At the same time, the proposed metrics still create comparability issues across different defense goals. Low facial recognition (FR) similarity is meaningful for FaceLock (and also relevant for EditShield), but inappropriate for edit-prevention methods that preserve identity by design like PhotoGuard. Lastly, we find the introduced CLIP-I (Radford et al. (2021)) score problematic as well. Even within FaceLock, CLIP-I can vary widely depending on whether identity removal occurs via heavy visual distortion or more subtle facial changes, yet both outcomes may represent equally valid defenses and CLIP-I will only support one.

Following this analysis, we find that EoDR FaceLock achieves largely the same results as vanilla FaceLock across most metrics, with the exception of FR, where it yields a modest improvement. Notably, the improvement appears specifically on the metric we argue is most meaningful for evaluating FaceLock's objective, indicating progress resulting from our extension. However, while EoDR improves FR relative to FaceLock, its performance still remains far from the EditShield baseline, indicating that further improvements to the FaceLock-like methods are necessary.

Now more broadly, these difficult comparisons highlight a misalignment between current evaluation practices and the diversity of defense objectives: edit-prevention methods should be evaluated primarily using prompt fidelity, while disruption defenses should instead be assessed via facial recognition measures. That said, both categories require an explicit analysis of the trade-off between defense effectiveness and corruption of the protected image in order to properly contextualize their capabilities. Motivated by these observations, we propose an additional evaluation framework that directly captures this trade-off.

### 4.1.1 Additional Evaluation Pipeline

To better illustrate the limitations of the evaluation pipeline in its current form, we introduce a deliberately flawed baseline, Simple FaceLock, which uses FaceLock's face recognition and alignment models to locate facial landmarks, expands a face bounding box by 25%, and replaces the corresponding region with uniform noise $\mathbf{U}(-1, 1)$. While this trivially removes biometric identity (Figure 2), it does so by destroying the face at publication time, making it impractical as a defense. Despite its obvious shortcomings, Simple FaceLock

still achieves strong scores on several metrics used in prior work (Table 4), thereby explicitly demonstrating the inadequacy of existing evaluations.

Figure 2: Comparison of defenses and edits on defended images for PhotoGuard, EditShield, FaceLock, and the ill-posed Simple FaceLock.

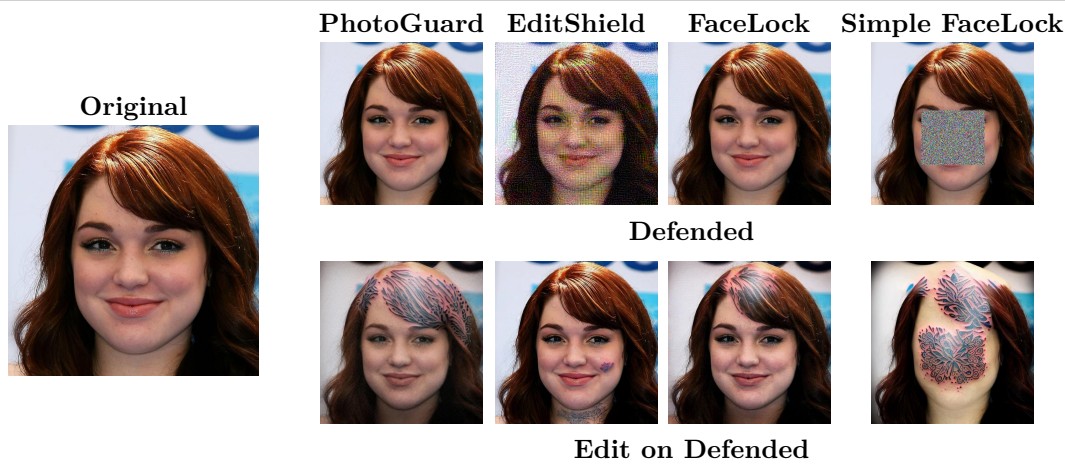

With this in mind, we propose additionally evaluating image integrity **prior to editing**, comparing the original image to its protected counterpart using PSNR, SSIM, LPIPS, and CLIP-I. The first three capture low-level visual similarity and are particularly informative when defenses apply subtle perturbations, allowing us to quantify how much a protection method interferes with the original image. The latter, in contrast, measures high-level semantic similarity and is most relevant to identify defenses that introduce substantial distortions that may destroy image content. In such cases, downstream edits probably become meaningless and biometric information may be removed the cost of no longer being able to publish the intended image.

We report the performance of all evaluated methods under our proposed additional evaluation pipeline in Table 5. Under this framework, FaceLock's behavior can be interpreted more clearly: although it achieves lower defense scores than EditShield in our reproducibility study, it substantially better preserves the original image at publication time, making the trade-off between protection and image degradation explicit. However, since PhotoGuard is more effective than FaceLock while also preserving image quality somewhat better, FaceLock does not appear to offer the strongest balance among the evaluated methods. This offers yet another perspective to our conclusion that FaceLock does not offer state-of-the-art identity protection in generative image editing.

Table 5: Quantitative evaluation on image integrity **prior to editing** (PSNR, SSIM, LPIPS and CLIP-I).

| | Visual Similarity | | | Semantic Similarity |
|---|---|---|---|---|
| Method | PSNR ↑ | SSIM ↑ | LPIPS ↓ | CLIP-I ↑ |
| PhotoGuard | **38.1329 ± 1.7246** | **0.9509 ± 0.0150** | **0.1333 ± 0.0214** | **0.9803 ± 0.0118** |
| EditShield | 34.4477 ± 0.9966 | 0.8683 ± 0.0181 | 0.3092 ± 0.0355 | 0.9607 ± 0.0146 |
| FaceLock | 38.0669 ± 1.6674 | 0.9490 ± 0.0141 | 0.1633 ± 0.0251 | 0.9775 ± 0.0111 |
| EoDR FaceLock | 38.0785 ± 1.6719 | 0.9496 ± 0.0140 | 0.1603 ± 0.0246 | 0.9780 ± 0.0110 |
| Simple FaceLock | 18.4689 ± 0.8984 | 0.8064 ± 0.0182 | 0.2405 ± 0.0212 | 0.6893 ± 0.0503 |

Regarding our extensions, EoDR FaceLock maintains the same image preservation performance as vanilla FaceLock, therefore, together with its higher FR score, it establishes itself as a consistent improvement over the original method. By contrast, Simple FaceLock performs poorly under our additional evaluation. Because our proposed framework ensures that defense effectiveness is not assessed in isolation, it successfully

captures this failure quantitatively, finally yielding numerical proof to dismiss the inherently ill-posed Simple FaceLock as a valid defense.

## 4.2 Qualitative Results

In our reproduction, we also observe substantial differences from the qualitative results reported in the original paper. To investigate these discrepancies, we systematically analyze edits on original versus protected images and contrast successful and failed cases, which additionally helps contextualizing the quantitative gaps. From an intuitive criterion for successful defense for FaceLock's facial biometric information removal objective, we find that for most prompts across the evaluated dataset, only a very small fraction of images are defended. We present a set of examples in Figure 3, including cases not highlighted in the original paper, to illustrate the dominant behaviors we observed in practice. We also provide a deeper dive into the examples in the appendix (Figures 5, 6, and 7), showcasing much more image-prompt pairs, spanning all prompt categories, and including some of the sporadic success scenarios, to make sure we enable a broad and fair overview of the method's results.

Figure 3: Qualitative comparison of image editing with and without FaceLock defense. Columns show the original image, edited (original) image, defended image, and edited defended image.

For most images, we observe little difference between edits applied to the original and protected images, and the defense typically fails to introduce changes substantial enough to remove biometric identity. For example, in Figure 3(a), the "pink hair" edit remains perceptually unchanged. In other cases, the defense mainly induces a global color shift rather than identity distortion, leaving facial identity intact and resembling

edit disruption (as in PhotoGuard) rather than FaceLock's intended mechanism. In the specific instance of Figure 3(b), the defended edit is even more visually altered than the original. That said, we've noticed slightly higher success rates for background-alteration prompts, where success often stems from blending the face into the edited background and thereby hiding it.

We observe a recurring link between apparent defense success and failures in the editing process itself. In some cases (Figure 3(c)), the edit on the original image already produces severe artifacts, making the defended output appear successful only because the edit is unstable rather than because FaceLock meaningfully intervenes. We also find cases where the edit on the original image is weak or fails, but the same edit is applied more strongly to the defended image without removing identity (Figure 3(d)), suggesting the defense can sometimes even amplify edits and produce the opposite of its intended effect. With all that in mind, we still observe successful cases that align with FaceLock's intended goal. For example, in Figure 3(e), the edit is applied as intended, but the subject's face is distorted to the point that the celebrity doesn't look the same and the image clearly appears AI-generated/edited.

We further conducted a reproducibility study (Figure 8) using the exact image–prompt pairs from Figure 6 in Wang et al. (2024), in order to test whether we could replicate their qualitative results. While we were unable to reproduce most of their examples, experiments across multiple random seeds revealed an important trend: even though FaceLock performs poorly overall in our evaluation, it often succeeds for *some* seed. In Figure 8, FaceLock succeeds for at least one seed in 6 out of 9 image–prompt pairs. Although this highlights the potential of the approach, it also demonstrates substantial sensitivity to randomness - which motivated our implementation of EoDR FaceLock.

Our experiments show that EoDR FaceLock almost always succeeds in the cases where vanilla FaceLock succeeds, while also extending success to a few additional scenarios, although the improvement is incremental rather than transformative. We further evaluated these additional cases across multiple editing seeds, focusing on examples where EoDR FaceLock succeeded but vanilla FaceLock failed under the initial seed. As illustrated in Figures 4 and 9, EoDR FaceLock also remained more consistent across tested seeds, ratifying the indication of a step towards improved robustness.

Figure 4: Comparison of EoDR FaceLock and vanilla FaceLock across different editing seeds. For the initial seed (42), EoDR provided protection while vanilla didn't. For seed 0, EoDR provided better protection than Vanilla, although not yet ideal. For seed 234, both methods proper (and similar) protection.

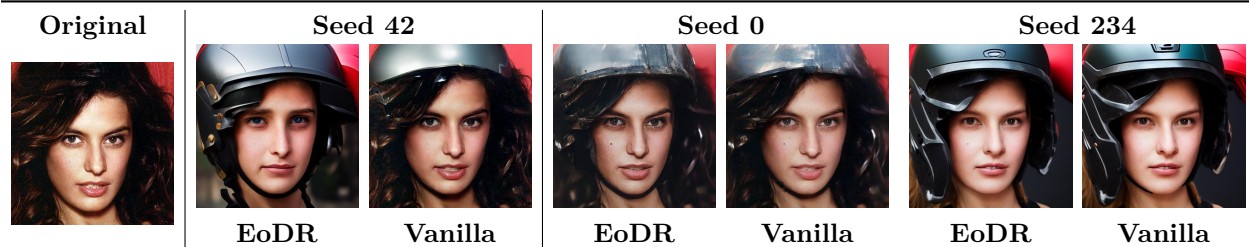

## 4.3 Generalization Experiments

Table 6 reports FaceLock's quantitative results on FaceForensics++ (Rössler et al. (2019)), which are broadly consistent with those obtained on CelebA-HQ (Karras et al. (2017)). While we observe improved prompt fidelity (CLIP-S and LPIPS), this is difficult to interpret given our earlier argument that it is not meaningful for FaceLock's objective. Similarly, the only statistically significant image integrity improvement is in CLIP-I, which we also consider an unreliable indicator. Overall, the results lead to the same conclusion: we cannot support the claim that FaceLock reliably removes biometric identity in edited outputs. On the other hand, FaceForensics++ is a more diverse and challenging dataset, suggesting that FaceLock's (limited) performance generalizes beyond CelebA-HQ, under more realistic conditions and combinations of purification methods in the wild. Figure 10 shows corresponding qualitative examples on FaceForensics++ across prompts from

the three categories discussed earlier, and the outcomes closely mirror the ones on CelebA-HQ once again, reinforcing FaceLock's limited reliability, but consistency.

Table 6: Quantitative evaluation on prompt fidelity for FaceLock on the FaceForensics++ dataset. All results are averaged over the 25 prompts for editing.

| | Prompt Fidelity | | | | Image Integrity | |
|---|---|---|---|---|---|---|
| Method | CLIP-S ↓ | PSNR ↓ | SSIM ↓ | LPIPS ↑ | CLIP-I ↓ | FR ↓ |
| FaceLock | $0.0720 \pm 0.0258$ | $24.0042 \pm 2.8763$ | $0.8490 \pm 0.0475$ | $0.3050 \pm 0.0400$ | $0.8159 \pm 0.0522$ | $0.8153 \pm 0.1378$ |

## 5 Conclusion

This work highlights both the relevance of biometric defenses against generative image editing and the conceptual potential of FaceLock (Wang et al. (2024)). The method builds upon general image defenses such as both PhotoGuard and EditShield pipelines (Choi et al. (2024); Chen et al. (2023)) and introduces a shift in perspective by reframing defense as identity distortion in edited outputs rather than edit suppression or complete distortion, what constitutes an interesting and promising direction for future research.

We also acknowledge the Authors' efforts toward reproducibility, including detailed experimental descriptions across the main paper and appendix, as well as the release of an open-source implementation. However, our study reveals substantial room for improvement. In particular, reproducibility is hindered by incomplete specification of dataset splits and random seeds, limited robustness of the codebase to different environments, and reliance on undocumented assumptions regarding image preprocessing. More critically, we identify several inconsistencies between the method as described in the paper and its implementation, as well as a lack of transparency regarding optimization targets, which together pose significant challenges for faithful reproduction.

Moreover, despite following the provided setup as closely as possible, we were unable to reproduce the performance trends reported in the original paper and therefore cannot confirm the claim that FaceLock achieves state-of-the-art performance relative to the proposed baselines. Qualitatively, we observed successful defenses directly attributable to the defense pipeline in only a small fraction of cases, in contrast to the uniformly strong examples presented in Wang et al. (2024). In many instances, apparent successes were instead correlated with failures or instabilities in the editing process itself.

Finally, our work highlights FaceLock's lack of robustness to editing seeds and proposes a promising extension that modestly improves the reliability of biometric distortion without interfering with image quality in publication time. Beyond these findings, our study underscores the need to reassess evaluation practices in this field. Different defense paradigms pursue fundamentally different objectives and should not be evaluated using the same set of metrics. We argue for the importance of contextualizing evaluation metrics according to defense goals and, most importantly, for explicitly measuring the trade-off between protection effectiveness and image distortion at publication time, which we propose in this work with our additional evaluation pipeline.

### 5.1 Limitations and Future Work

Although we consider this a comprehensive reproducibility study, examining both quantitative and qualitative aspects of the main claims in the FaceLock paper (Wang et al. (2024)) as well as the surrounding baseline methods, our work still has several limitations and naturally motivates future work. We did not systematically explore hyperparameters for either the defense or editing pipelines, focusing instead on the configurations reported by the authors. Likewise, we did not comprehensively test architectural alternatives included in the codebase, primarily running the setup emphasized as best-performing in the paper: InstructPix2Pix (Brooks et al. (2022)) for editing and the VGG-based backbone (Simonyan & Zisserman (2015)) for LPIPS feature extraction (Zhang et al. (2018)), rather than alternatives such as Stable Diffusion image-to-image (Meng et al. (2022)) or AlexNet-/SqueezeNet-based LPIPS backbones (Krizhevsky et al.

(2012); Iandola et al. (2016)). We also did not reproduce all auxiliary experiments from the original paper, such as the full ablation studies.

Despite these limitations, we believe our study remains valuable by reassessing the method in depth, documenting reproducibility issues, and contributing an extension to FaceLock and a proposal for a better evaluation pipeline. For future work, reproducibility efforts could fill the remaining gaps by systematically testing hyperparameter and architectural sensitivity, and completing reproduction of the remaining auxiliary experiments. We also believe that our robustness-oriented extension, EoDR FaceLock, could be further investigated and improved through more hyperparameter tuning or by incorporating new strategies to better handle the stochasticity inherent in diffusion-based editing. More broadly, we encourage future work in the field to incorporate our proposed evaluation framework, which explicitly captures the trade-off between biometric protection effectiveness and distortion of the protected image at publication time, enabling more principled comparisons across methods with fundamentally different defense objectives.

## 6 Broader Impacts and Ethics Statement

Recent advances in generative editing pose significant risks to individual privacy and biometric security. This work contributes to the ethical development of safeguards by critically auditing existing defenses, as unreliable protection may offer users a false sense of security. By highlighting the limitations of state-of-the-art methods, specifically the discrepancies between claimed and actual performance in FaceLock (Wang et al. (2024)), we advocate for greater transparency and reproducibility in this research field. While our proposed EoDR extension offers incremental gains in robustness for FaceLock, we emphasize that current defenses are not yet a complete solution for unauthorized manipulation. Another important ethical contribution of this study lies in our argument for defense-objective contextualized evaluation and our proposed additional evaluation framework, which mandates a transparent accounting of the trade-off between image integrity prior to editing and defense effectiveness. We believe this shift is essential for developing responsible technologies that genuinely protect individuals against the misuse of their digital likeness.

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

# A    Additional Qualitative Results

The figures below present more qualitative results for FaceLock, PhotoGuard and EditShield (Wang et al. (2024); Choi et al. (2024); Chen et al. (2023)) on specific cases of image editing on Celeba-HQ images (Karras et al. (2017)), spanning the three categories reported by the Authors: modification of facial attributes, accessory addition and background modification. The results are separated into the original image, protected images after applying the defense pipelines, and finally the edits on both the original image and the protected images.

## A.1    Qualitative Results on Facial Feature Modification

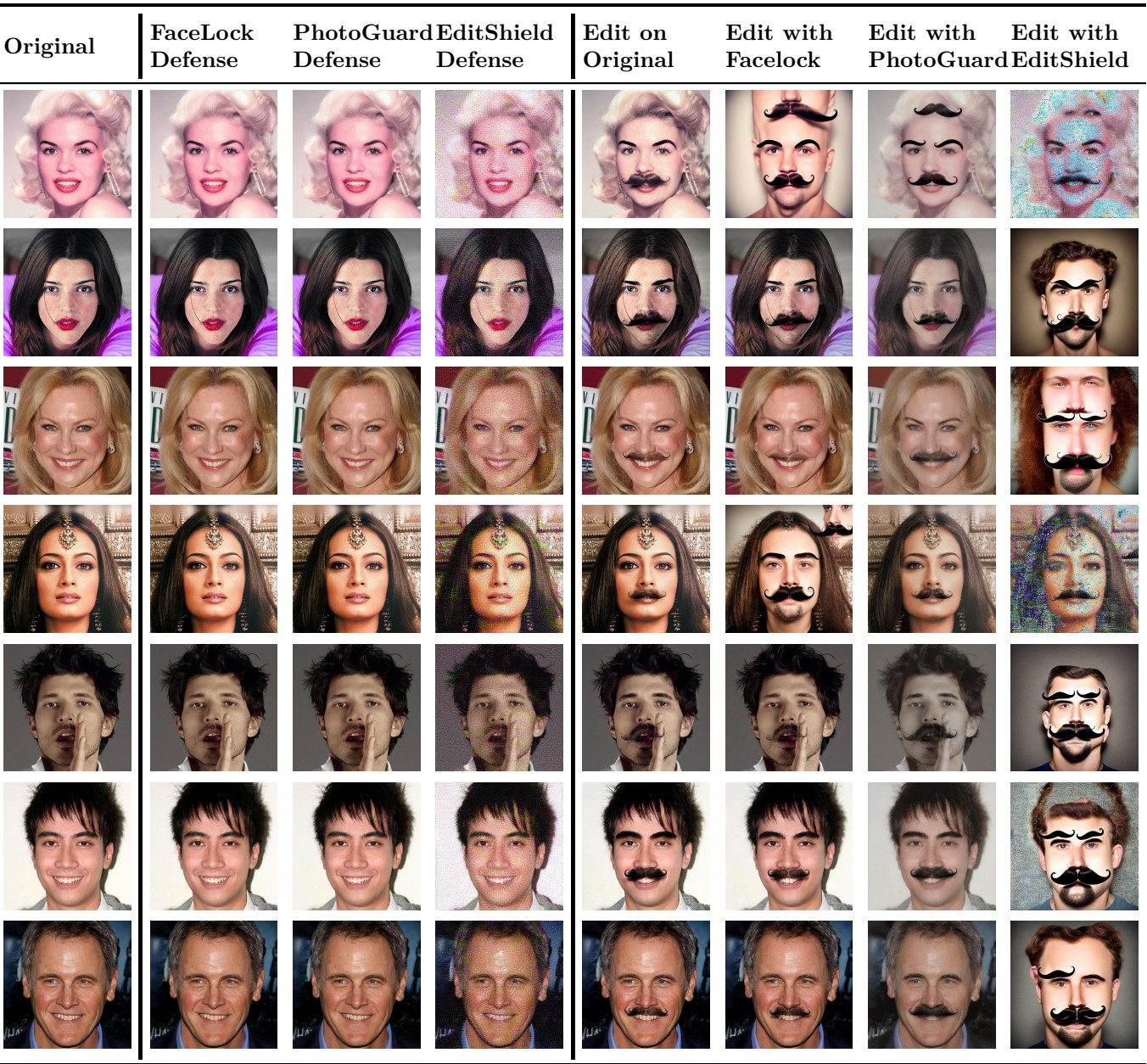

Figure 5: **Prompt: "Let the person grow a mustach"**

## A.2 Qualitative Results on Accessory Addition

| Original | FaceLock Defense | PhotoGuard Defense | EditShield Defense | Edit on Original | Edit with Facelock | Edit with PhotoGuard | Edit with EditShield |
|----------|------------------|--------------------|--------------------|------------------|--------------------|----------------------|----------------------|

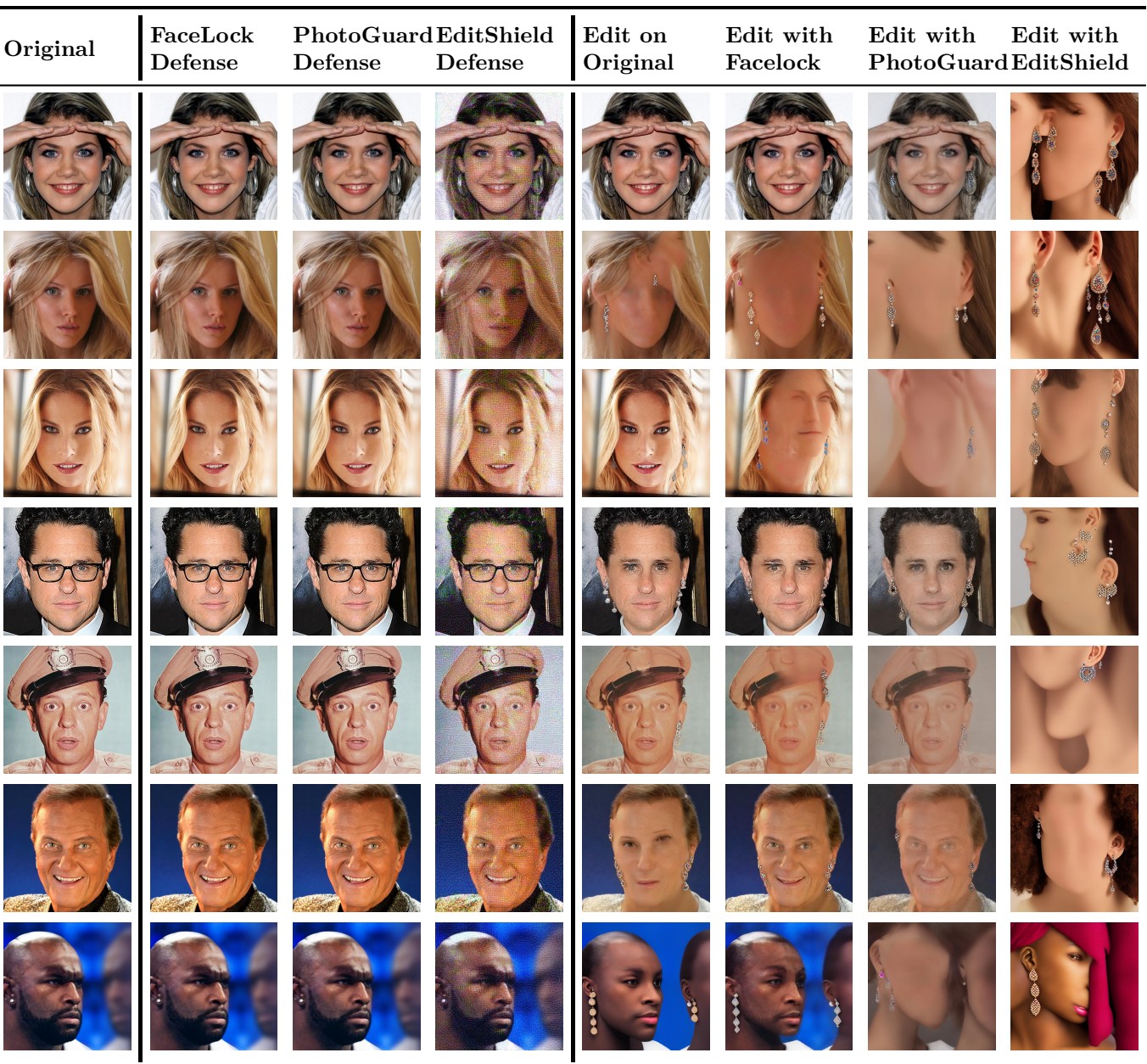

Figure 6: **Prompt: "Let the person wear earrings"**

## A.3   Qualitative Results on Background Modification

| Original | FaceLock Defense | PhotoGuard Defense | EditShield Defense | Edit on Original | Edit with Facelock | Edit with PhotoGuard | Edit with EditShield |
|---|---|---|---|---|---|---|---|

Figure 7: **Prompt: "Add a city skyline background"**

## B Reproducibility of Wang et al. (2024) Figure 6

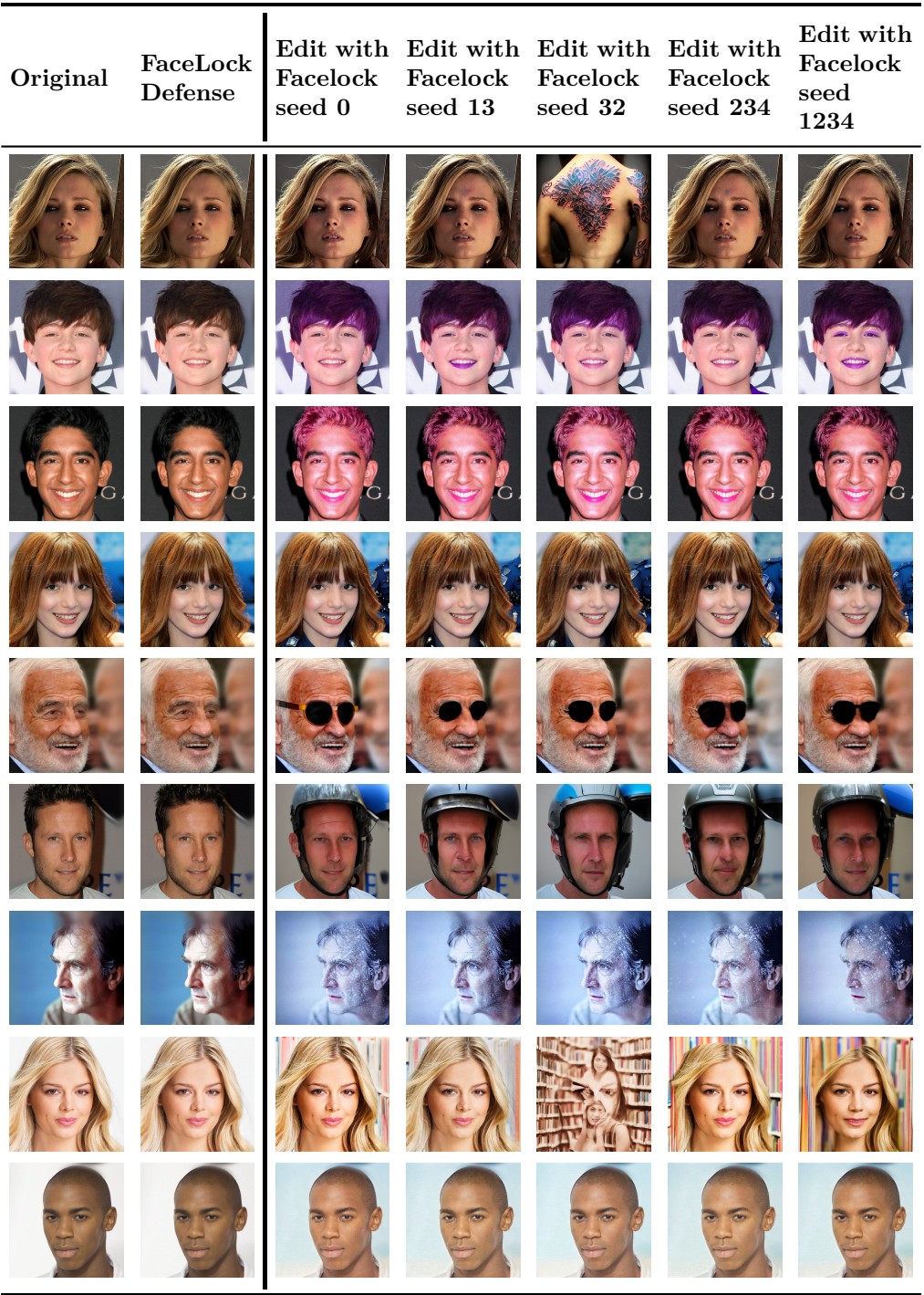

Figure 8: FaceLock on images of Wang et al. (2024) Figure 6 for multiple seeds. From top to bottom, prompts are: 1. "Let the person have a tattoo"; 2. "Let the person wear purple makeup"; 3. "Turn the person's hair pink"; 4. "Let the person wear a police suit"; 5. "Let it be snowy"; 6. "Set the background in a library"; 7. "Change the background to a beach."

## C   Additional Qualitative Results for EoDR FaceLock

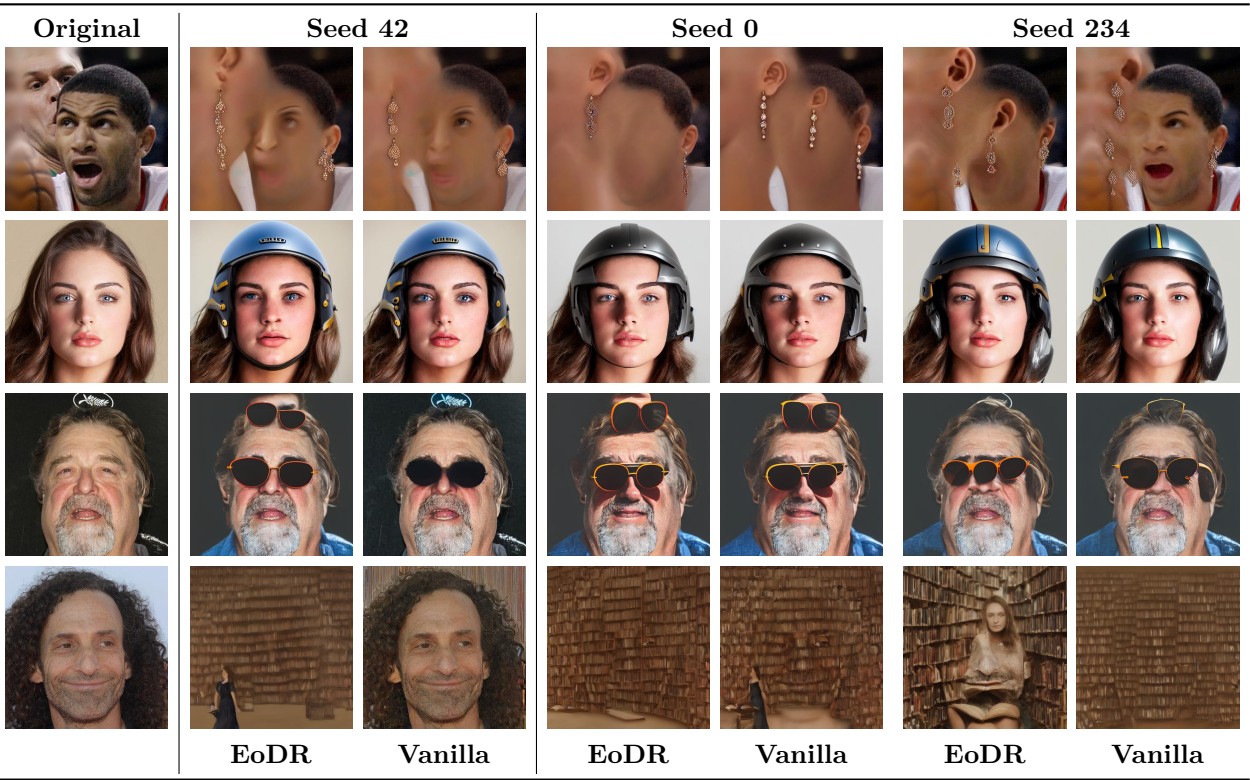

Figure 9: Comparisons of EoDR FaceLock and vanilla FaceLock for a few image-prompt pairs for which EoDR provided more defense effectiveness, more stability across editing seeds, or both.

## D Qualitative Results of Generalization Experiment

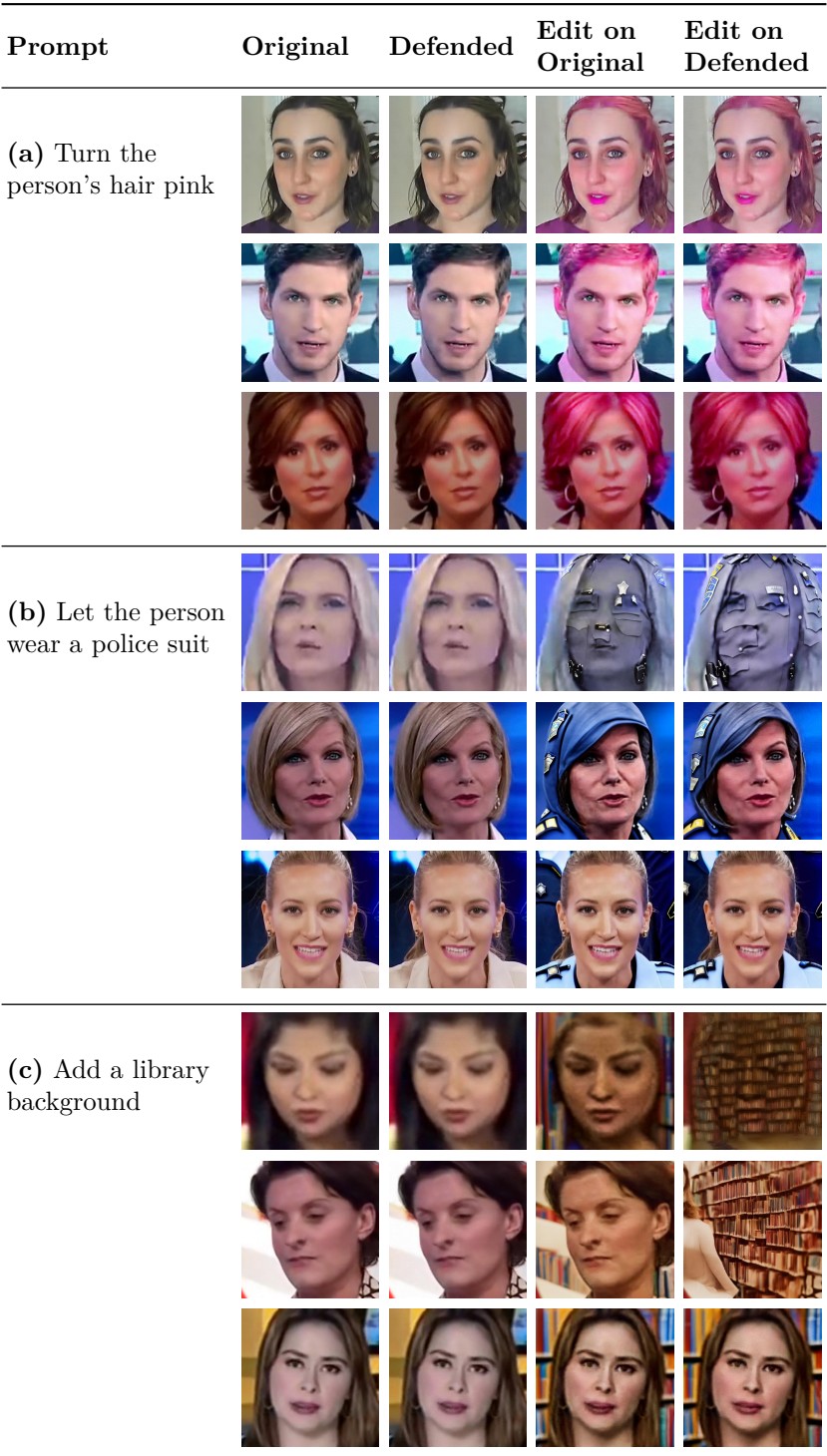

Figure 10: Qualitative results for FaceLock on FaceForensics++ dataset (Rössler et al. (2019)). Columns show the original image, defended image, edit on the original image, and edit on the defended image. Each prompt is shown for three different source portraits.

