# OpenReview forum: "Revisiting "Edit Away and My Face Will not Stay: Personal Biometric Defense against Malicious Generative Editing""
_TMLR — Accepted by TMLR_

### Review · Reviewer_Rpgs · 2026-02-23

**Summary Of Contributions:**

The paper revisits a prior work on biometric protection against malicious generative editing and conducts a comprehensive reproducibility study of the associated method: FaceLock. Through a set of experiments, the paper reveals several mismatches between the reported and reproduced performance, showing that FaceLock is not only sensitive to input seeds but also unstable across edit prompts/datasets.

**Audience:**

Yes

**Audience Explanation:**

The paper studies a timely and important topic in biometric protection against malicious generative editing, a critical threat posed by deepfakes. While the paper does not invent new methods, the reproducibility results and the performance gap it reveals would be of interest to TMLR’s audience who have read the FaceLock paper or are working on a similar research topic.

From a broader perspective, however, it is unclear whether the specific reproducibility study presented in this work contributes clear, actionable implications for advancing the state-of-the-art literature on biometric protection against malicious generative editing. In my opinion, because of its limited scope and lack of mechanistic insights, the paper may not be interesting to a broader audience in trustworthy ML communities.

**Broader Impact Concerns:**

The paper conducts a reproducibility study on a prior protection method. While no direct ethical concerns are relevant, the paper would benefit from a discussion of the technology's potential ethical implications, particularly the risk of dual use.

**Claims And Evidence:**

Yes

**Claims Explanation:**

The paper's main contribution arguments are supported by reproducing the experimental results from a prior paper. Sufficient details and discussions are given. From a reproducibility perspective, the paper makes a commendable effort to document the performance gap in a rigorous and transparent manner, which may inspire future research looking into the issues. While the reproducibility study is supported by clear documentation and comprehensive empirical results, the paper does not sufficiently explain why the gap occurs. From my perspective, the paper falls short of providing deeper mechanistic insights, which is a crucial limitation that weakens its scientific contributions.

For instance, the paper emphasizes that FaceLock's design aims to disrupt biometric identity, a distinctive feature that differs from prior methods (PhotoGuard and EditShield), which should intuitively enhance image integrity metrics. However, the reproducibility study reveals a contradictory observation: FaceLock is even worse than prior methods on image integrity metrics, such as CLIP-I and FR. This seems surprising, but what are the underlying reasons? It is unclear whether the issue points to transparency issues in the original implementation or to a methodological weakness in the FaceLock design.

**Requested Changes:**

To strengthen the work, the paper is recommended to delve deeper into the insights that explain the revealed performance gap and to provide more discussion of their implications for improving the current state of protection research on malicious generative editing. In addition, the paper would benefit from broadening the scope of its reproducibility study and discussions to encompass more existing protection methods and investigate whether the performance drop is unique to FaceLock or is generally observed across the literature.

---

> ### Author Response · Authors · 2026-02-24
> **Addressing questions, misunderstandings and requests from Review 1.**
>
> Before anything, thank you for your valuable feedback. Important points were raised and, regardless of potential small disgagreements, we genuinely appreciate your suggestions for strengthening the manuscript. In the revised version, we will expand our discussion to provide deeper and more explicit insights that help explain the observed performance gaps. We will also include a dedicated discussion of potential ethical implications to further contextualize the work and clarify its broader impact.
>
> Noving moving forwards, trying to address potential misunderstandings and questions, we provide answers to the comments on both the TMLR evaluation criteria.
>
> Concerning points raised on TMLR criterion 1, we understand the importance of providing mechanistic insights and acknowledge that this aspect may not have been explored as thoroughly as it could have been. Regarding the specific example raised, we would like to clarify two points:
>
> 1. First, as discussed in Section 4.1 (page 8, paragraph 3, starting with "At the same time"), we do not expect FaceLock to perform strongly on the CLIP-I image integrity metric. Even within sucessful FaceLock cases, CLIP-I scores can vary significantly depending on whether identity removal is achieved through strong visual distortion or through more subtle facial modifications. Both outcomes may constitute equally valid defenses, yet CLIP-I inherently favors only one of these. Therefore, lower performance on this metric should not be surprising.
>
> 2. Second, we attribute the lower FR scores to FaceLock’s large number of complete defense failures, which stem from its sensitivity to the diffusion process randomness. We explicitly highlight this variability as a methodological weakness. Our proposed extension, EoT FaceLock, is motivated by this issue and aims to mitigate seed sensitivity by accounting for diffusion variability. We acknowledge, however, that this connection could have been stated more explicitly in the quantitative results section, and we will revise the manuscript to clarify this point.
>
>
> Concerning points raised on TMLR criterion 2, while we agree that our work does not introduce a fundamentally new paradigm, we believe it makes meaningful contributions with actionable implications for the field:
>
> 1. First, we introduce EoT FaceLock, a variation of FaceLock designed to address seed variability. Our experiments show incremental improvements in robustness, though we are careful to state that this remains an incomplete solution. We view it as a step toward a broader research direction focused on improving stability under diffusion randomness.
>
> 2. Second, we propose an extended evaluation pipeline that explicitly measures the trade-off between protection strength and image distortion prior to editing. We argue that current evaluation protocols insufficiently capture this trade-off. In particular, defenses that strongly prevent editing often do so at the cost of significant visual distortion, yet this tension is not typically quantified. To illustrate this limitation, we even implement a deliberately ill-posed pipeline which, under standard evaluation metrics alone, outperforms all other evaluated methods on most metrics. However, when assessed under our proposed evaluation framework, which accounts for distortion relative to the original image, its shortcomings become clear. This example highlights a central claim of ours that future work should incorporate trade-off-aware evaluation protocols to avoid favoring methods that achieve protection through excessive degradation.

---

> ### Comment · Reviewer_Rpgs · 2026-03-06
>
> I thank the authors for the clarifications. I had two follow-up comments specific to the following statements. Please clarify if necessary.
>
> > Second, we attribute the lower FR scores to FaceLock’s large number of complete defense failures, which stem from its sensitivity to the diffusion process randomness. We explicitly highlight this variability as a methodological weakness. Our proposed extension, EoT FaceLock, is motivated by this issue and aims to mitigate seed sensitivity by accounting for diffusion variability.
>
> I looked into the result tables and the subsections regarding EoT FaceLock. What I understand is that EOT FaceLock consistently performs worse than PhotoGuard across all metrics (including FR). The improvement over the original FaceLock is only marginal in terms of FR in Table 4, while for all the remaining metrics, either there is no improvement or the increase in the averaged statistics is within the standard deviation. The above comparison results do not seem to support the implication that Facelock's failure is due to random diffusion and that the proposed EoT FaceLock can resolve the issue.
>
> > To illustrate this limitation, we even implement a deliberately ill-posed pipeline which, under standard evaluation metrics alone, outperforms all other evaluated methods on most metrics. However, when assessed under our proposed evaluation framework, which accounts for distortion relative to the original image, its shortcomings become clear.
>
> This is not surprising. If most of the face is blocked (as shown in Simple FaceLock), the identity cannot be preserved because essential identity information is completely lost. If the goal is to demonstrate the trade-off, it should follow a similar utility constraint on the extent to which the input images can be perturbed. Otherwise, one can also argue the same using more extreme cases: completely blocking the whole image, or replacing it with someone else's.

---

> > ### Author Response · Authors · 2026-03-06
> > **Answering follow-up comments.**
> >
> > Thank you very much for your follow-up comments. We hope we can address all the points you raised:
> >
> > 1. EoT FaceLock indeed performs worse than PhotoGuard across all metrics, and we did not intend to suggest otherwise. Our goal was simply to show that it represents a modest improvement over the original FaceLock. In particular, EoT FaceLock achieves small but consistent gains (remember that results are averaged over 25 prompts from multiple categories) in the FR score. In the paper, we argue that FR is the only metric that meaningfully reflects the effectiveness of FaceLock-type defenses. Prompt-fidelity and CLIP-I scores are less informative in this context: effective FaceLock defenses can exhibit either high or low values for these other metrics.
> >
> > Now regarding seed sensitivity, we do observe that FaceLock suffers from it: for a given image, the method often succeeds for some seeds but fails for others. Our argument is therefore not that FaceLock fundamentally fails, but rather that it lacks consistency. The consistent improvements in FR score - computed by averaging results across prompts - together with the qualitative examples we provide, suggest that EoT FaceLock slightly improves robustness to seed choice. Importantly, we do not claim that it fully resolves this issue. As stated in the paper, EoT FaceLock generally succeeds in the same scenarios as vanilla FaceLock, with a small number of additional successes. These additional successes modestly improve robustness across seeds, which explains the small but consistent gains in FR score. Hence, our goal was not to propose a new state-of-the-art defense, but rather to take a small step toward stabilizing FaceLock, which introduced a promising and conceptually interesting approach to identity preservation in generative image editing.
> >
> >
> > 2. We believe this comment actually reflects the exact point we intended to make. We do not claim that Simple FaceLock is a good or practical defense. It clearly is not. Its purpose is only to illustrate the importance of quantitatively accounting for image preservation when evaluating defenses. In other words, the contribution here is not Simple FaceLock itself, but the additional evaluation pipeline we propose to measure the trade-off between defense strength and image preservation.
> >
> > To illustrate this point more clearly, forget about Simple FaceLock and consider the argument we made for EditShield on the paper. Apart from Simple FaceLock, EditShield performs best across almost all metrics - particularly in FR score. Based solely on these metrics, one might conclude that EditShield is the best method and should simply be adopted. However, when we evaluate the same methods using the additional pipeline we propose, a different picture emerges. The stronger defense provided by EditShield comes at a substantial cost: it significantly degrades the image prior to editing. This makes the trade-off explicit. One could choose a stronger defense that noticeably degrades the image at publication time (EditShield), or opt for slightly weaker defenses that preserve the image almost perfectly before editing (PhotoGuard for state-of-the-art, or FaceLock). And there is no universally correct choice, but our evaluation framework allows this trade-off to be considered explicitly rather than implicitly assuming that EditShield is strictly superior.

---

### Review · Reviewer_xD2G · 2026-02-24

**Summary Of Contributions:**

The authors conducted a three-pronged evaluation—technical, quantitative, and qualitative—to see if FaceLock's results hold up under independent testing. The study
reveals several "undocumented" features in the FaceLock codebase, such as a staged optimization strategy and the fact that the method optimizes directly for the
evaluation metric (LPIPS), which was not disclosed in the original paper. To address FaceLock's extreme sensitivity to random seeds in diffusion models, the authors
proposed an extension called Expectation over Transformation (EoT) FaceLock. This version simulates diffusion randomness during the defense process to create
more robust protection.

Key Strengths
1. Constructive Skepticism: Rather than just pointing out failures, the authors provide a technical explanation for why the original results might have appeared better than they were (e.g., the hidden optimization for LPIPS).
2. Iterative Improvement: The authors implemented EoT FaceLock which is a small iterative improvement over FaceLock, addressing the susceptibility of the model to random seeds.
3. Testing on Difficult Dataset: By expanding to the FaceForensics++ dataset, the study provides a more "real-world" assessment of how these biometric defenses might behave in the wild.
4. Clarity on Metrics: The paper identifies a fundamental flaw in the field: researchers are often using the wrong metrics to measure success, leading to misleading comparisons between different "types" of AI defenses.

Key Weaknesses
1. Experiment on a subset: The authors performed all the experiments on a subset of the dataset used in FaceLock paper. Comparison of different models on a subset  may not always match with the comparison on the full dataset.
2. Computational Constraints: Due to hardware limitations (40GB A100 GPU), the authors could not run the pipeline at the original $1024 \times 1024$ resolution,  potentially impacting the absolute performance numbers compared to the original study.

**Audience:**

Yes

**Audience Explanation:**

The paper's findings would appeal to individuals in the following areas:

1. AI Safety and Privacy Researchers

Biometric Defense: The paper evaluates FaceLock, a novel method designed to protect personal facial images from "malicious generative editing".
Vulnerability Analysis: It exposes critical vulnerabilities in current biometric defenses, such as extreme sensitivity to the random seeds used in diffusion models.
Privacy Protection: For those focused on identity security, the paper discusses how to prevent unauthorized transformations of personal images while still allowing them to be shared online.

2. Reproducibility and Rigor Specialists

Code-to-Paper Discrepancies: The study identifies significant mismatches between FaceLock’s theoretical description and its actual implementation, such as undocumented staged optimization
strategies.
Benchmark Validation: The authors found they could not replicate the "state-of-the-art" performance trends reported in the original FaceLock paper.
Generalization Testing: The paper tests the defense on a more challenging, real-world dataset (FaceForensics++) to see if the laboratory results hold up under more realistic conditions like
motion blur and compression.

3. Evaluation and Metrics Experts

Metric Critique: The authors argue that standard metrics like "prompt fidelity" are fundamentally ill-suited for evaluating biometric defenses.
New Evaluation Framework: They propose a more robust evaluation pipeline that explicitly measures the trade-off between how well an image is protected and how much the original image is
degraded by the defense.

4. Generative Model Developers

Diffusion Robustness: The introduction of EoT FaceLock provides a technical extension that improves the robustness of defenses against the inherent randomness (stochasticity) of
diffusion-based editing.

**Claims And Evidence:**

No

**Claims Explanation:**

The qualitative analysis presented in the paper clearly identifies many failure cases of FaceLock. The shortcomings
of multiple evaluation metrics for comparing different defense models were also clearly identified.


However, the quantitative analysis part of the paper is not comprehensive enough to justify the claim. The authors
ran the experiments on a subset of the dataset used in FaceLock paper. From such experiments, it cannot be said
"without any reasonable doubt" that FaceLock consistently performs worse than EditShield and PhotoGuard.

**Requested Changes:**

The authors ran experiments on 1000 randomly selected images from CelebA-HQ validation split. It may happen
that these 1000 images were cherry-picked from the whole set (i.e. 2000 images) to prove FaceLock inferior
compared to EditShield and PhotoGuard. Therefore, to actually reproduce the study of FaceLock paper, it is
recommended that the authors perform the experiments on the whole CelebA-HQ validation split and report the
results.

---

> ### Author Response · Authors · 2026-02-25
> **Adressing questions, misunderstandings and requests from Review 2**
>
> Thank you for your thoughtful review. We sincerely appreciate both your recognition of our work’s contributions and your constructive feedback. We address the raised concerns point by point below:
>
> 1. Experimental resolution: We apologize if our explanation was not sufficiently clear. The authors of the FaceLock paper report conducting their experiments at a resolution of 512x512 as well. Our concern, however, is that the dataset they provide is in 1024x1024 resolution, while their released code does not include an explicit resizing operation. The only mention of the 512x512 setting appears in a single table in the appendix. This suggests that resizing was likely performed locally during their experiments but was not properly documented or incorporated into the shared pipeline. For completeness, we also attempted to run their pipeline directly on the 1024x1024 images. This was not feasible, not due to hardware limitations, but because the implementation operates entirely in floating-point 16 precision, which leads to numerical overflow at the higher resolution, and it is not easily adjustable to floating-point 32/64 precisions. We emphasize that this is not a limitation of our experimental setup. Rather, we mention it to clarify that the released pipeline is effectively designed to operate on 512x512 images, even though we believe this constraint is not made explicit enough in the original paper or codebase.
>
>
> 2. Sample size: During our experimental phase, we evaluated the pipeline on multiple randomly selected subsets of increasing size: 100, 400, and 1,000 images. The results were fairly consistent across these subsets. We naturally chose to report the results on the largest subset, as it provides the most statistically informative evaluation among the tested configurations. But considering the consistency we got from all our subset experiments and the randomness of the sampled images, we have no reason to believe that the observed results were biased because of the insufficient sample size. That said, we recognize the value of extending the evaluation to the full validation dataset, and we will include these expanded experiments in the revised version of the manuscript to further strengthen our conclusions.

---

### Review · Reviewer_SQJN · 2026-03-06

**Summary Of Contributions:**

The paper performs a reproducibility study of the FaceLock technique for defending face images against generative editing attacks; this earlier technique was proposed in a CVPR paper from 2024.

At a high level, the FaceLock paper applies a l-infty bounded adversarial perturbation to a given image; this perturbation is chosen such that any diffusion-based editing operation applied to this (new) image makes the biometric identity loss (a classification loss that measures similarity using features from a pretrained face recognition model) very high.

The authors attempt to reproduce the results of that paper and find that:
* FaceLock is susceptible to seed choice
* the published codebase is unstable
* the published code does not comport to the equations in the paper; in fact, the published code essentially resembles a previous technique (EditShield) as opposed to the new math proposed in the paper.
* the method does not generalize very well beyond CelebA-HQ.
These are backed up by various experimental results.

**Audience:**

Yes

**Audience Explanation:**

While the application focus is on biometrics, I believe that the robustness analyses carried out in the paper will be of interest to a significant  number of TMLR readers.

**Broader Impact Concerns:**

No concerns -- although since the broad topic relates to facial biometrics and identity protection, I would suggest that the authors please consider including a short broader impacts or ethics statement.

**Claims And Evidence:**

No

**Claims Explanation:**

* Convincingness: The paper overall seems to be a good reproducibility study, showing many limitations of the original paper. After reading it though, I came away unsatisfied. My sense is that the claims made in this paper (or rather, the conclusions drawn from the study) are not clearly spelt out. I understand why this might be the case --- essentially, the subtext appears to be that the original paper contains significant discrepancies, and perhaps the authors do not wish to make bold claims of this sort. But I suggest reconsidering this strategy and making clear statements (backed of course by equally clear evidence, and communicated in a respectful manner) based on their findings. It appears that the results of the paper may have enough for this, and the community might benefit if these claims are more prominent.

* Accuracy: Minor discrepancies in the eval procedure that I didn't fully understand, please clarify. Why only a 40GB GPU for evals? Why only 1000 images out of a subset of 2000?

* Clarity: while generally fine, the writing can be improved in a few places; see requested changes below.

**Requested Changes:**

- major: a clear summary of the main findings of the paper (perhaps a list of conclusions drawn from the reproducibility study) in the introduction would go a long way in making the paper more impactful.
- minor: The "EoT" suffix for EoT FaceLock is misleading, consider renaming to something else.
- minor: not sure if Section 2 really needs to be that long.
- minor: not sure if the appendix needs to be that long.
- minor: consider including an ethics statement.

---

> ### Author Response · Authors · 2026-03-06
> **Adressing questions, misunderstandings and requests from Review 3**
>
> Thank you for your comments and valuable feedback. We will take them into consideration in the revised version of the paper. Below we address some of the specific points you raised:
>
> 1. We believed our conclusions were sufficiently clear, but now that you pointed out that we migh have been a bit too cautious, we will revise the paper to state them more explicitly.
>
> 2. The mention of the 40GB GPU was intended only to illustrate a practical issue encountered when running the pipeline. However, since this point appears to have caused confusion (another reviewer also misunderstood the section), we will remove this reference and rewrite the section to clarify the point we were trying to make.
>
> 3. We experimented with multiple dataset sizes, increasing from 100 images up to 1000. The decision to stop at 1000 was primarily due to computational budget constraints. We have since resolved this issue and have just yesterday completed experiments on the full 2000 images. We will report these results in the revised version, although we can already say that the updated numbers barely differ from those previously reported.
>
> Regarding the requested changes, we very much appreciate them and will address them in the revised paper. The only possible exception is the appendix length, as we intentionally included an extensive set of experiments to allow for deeper qualitative analysis and comparisons with the original FaceLock paper.

---

### Author Response · Authors · 2026-03-11
**Summary of Revision.**

Dear Reviewers,

We would like to express our gratitude for your valuable feedback once again. We have carefully considered all comments and have incorporated a lot of your suggestions into the revised version of our manuscript. To facilitate the review process, we have summarized the primary modifications below, indicating which reviewer concerns each change addresses:

1. Introduction Refinement: We have restructured the last paragraph of the introduction to provide a more clear summary of our findings. (Reviewer 3)

2. Scope of Reproducibility: We revised this section to make it shorter, yet more informative and specific regarding our experimental design. It is meant to be a natural complement to the introduction. (Reviewer 3)

3. Full Dataset Experiments: We have updated all experimental results to reflect testing on the full validation set (2000 images), completely aligning with the original FaceLock experiments now. (Reviewers 2 and 3)

4. Discussion of Results: Both the quantitative and qualitative sections have been refined to make our claims and reasoning more explicit and robust. (Reviewers 1 and 3)

5. Conclusion Structure: We have moved "Limitations and Future Work" into a subsection inside the "Conclusion" section. (This was our own choice, not reviewer specific)

6. Broader Impacts and Ethics Statement: We added a new section briefly discussing the societal implications of generative editing, the current state of the generative editing protection, and both the ethical context and contributions of our reproducibility study. (Reviewers 1 and 3)

7. Terminology Update: We replaced the prefix "EoT" with the more precise "EoDR" (Expectation over Diffusion Randomness) to better reflect the technical nature of our extension. (Reviewer 3)

8. Quantitative Results Table: We relocated the "Simple FaceLock" method to a separate row and excluded it from the numerical rankings. As this baseline is a deliberately ill-posed method intended only to highlight flaws in current evaluation protocols, its exclusion allows for a more meaningful comparison between the four primary methods. We have updated the table caption to clarify this distinction for the reader. (This was our own choice, not reviewer specific)

We hope these revisions strengthen the paper. Thank you again for your time and for helping us improve this work.

Kind regards,
the authors

---

### Decision · Action_Editor_Z4R6 · 2026-04-15

**Recommendation:** Accept with minor revision

**Additional Comments:**

Before the final archival version is approved, the authors should ensure:

- Consistency in Terminology. Verify that the transition from "EoT" to "EoDR" (Expectation over Diffusion Randomness) is uniform throughout the text to avoid confusion with standard adversarial literature.

- Summary of Findings. Ensure the Introduction explicitly lists the primary conclusions of the reproducibility study (as requested by one reviewer) to improve the clarity.

- Distortion Context. Ensure the discussion of the "Simple FaceLock" baseline continues to be framed strictly as an evaluation tool rather than a proposed defense method.

**Reason for Certification.** The submission deserves the Reproducibility Certification. It goes beyond a simple analysis by identifying specific (undocumented) implementation details that led to the original paper's reported success.

**Audience:**

Yes

**Audience Explanation:**

The findings are of significant interest to researchers in AI safety, biometric security, and generative modeling. The paper highlights a critical issue in the field: the extreme sensitivity of adversarial defenses to the inherent stochasticity of diffusion models. In addition, the authors critique of existing evaluation metrics, demonstrated through the Simple FaceLock counter-example, provides a valuable insight for future work in this domain.

**Claims And Evidence:**

Yes

**Claims Explanation:**

The work provides a rigorous and technically sound reproducibility study of the "FaceLock" defense. Initially, reviewers raised valid concerns regarding the experimental scale and minor ambiguities in hardware constraints. However, the authors successfully addressed these in their revision by:

- Running all experiments on the full validation set, which confirmed their initial findings and eliminated potential sampling bias.

- Providing a detailed technical breakdown of the discrepancies between the original paper’s equations and the released codebase (e.g., undocumented staged optimization and LPIPS-specific tuning).

- Clarifying that numerical stability issues, rather than hardware limitations, were the primary barrier to higher-resolution testing.

While one reviewer remained skeptical regarding the mechanistic insights of the performance gap, the authors evidence regarding seed sensitivity and metric misalignment (the trade-off between protection and image integrity) constitutes a clear and accurate technical contribution within the scope of a reproducibility study.

---

> ### Author Response · Authors · 2026-04-26
> **Camera Ready Revision.**
>
> Hello, Mauricio. Thank you very much for the acceptance and also for your valuable revision requests. We've updated the paper with the following modifications:
>
> 1. We checked the consistency of the terminology and changed all the reference names that were still EoT to EoDR.
>
> 2. We have restructured the last paragraph of the introduction one more time to provide a more clear and explicitly listed summary of our contributions and findings.
>
> 3. We made some small modifications to image and table captions, as well as to the text in Quantitative Results to make it more clear that Simple FaceLock was not an actual baseline, but rather just a ill-posed method introduced to demonstrate some deficiencies of the current evaluation pipeline.
>
> Following the instructions we received by email, we have uploaded a Camera Ready Revision with the aforementioned modifications. We hope they properly attend your requests, but let us know if they don'y so we can adjust the paper further. Thank you once again.